# Evaluation of Concomitant Use of Anticancer Drugs and Herbal Products: From Interactions to Synergic Activity

**DOI:** 10.3390/cancers14215203

**Published:** 2022-10-23

**Authors:** Massimiliano Berretta, Lissandra Dal Lago, Mattia Tinazzi, Andrea Ronchi, Gaspare La Rocca, Liliana Montella, Raffaele Di Francia, Bianca Arianna Facchini, Alessia Bignucolo, Monica Montopoli

**Affiliations:** 1Department of Clinical and Experimental Medicine, University of Messina, 98122 Messina, Italy; 2Integrative Medicine Research Group (IMRG), Noceto, 43015 Parma, Italy; 3Department of Medicine, Institut Jules Bordet, Université Libre de Bruxelles (ULB), 1070 Brussels, Belgium; 4Department of Pharmaceutical and Pharmacological Sciences, University of Padova, 35123 Padova, Italy; 5Pathology Unit, Department of Mental Health, Physic and Preventive Medicine University of Campania “Luigi Vanvitelli”, 80131 Naples, Italy; 6Cancer Biology and Genetics Program, Memorial Sloan Kettering Cancer Center, New York, NY 10065, USA; 7Division of Medical Oncology, “Santa Maria delle Grazie” Hospital, ASL Napoli 2 NORD-, 80078 Pozzuoli, Italy; 8Gruppo Oncologico Ricercatori Italiani (GORI-ONLUS), 33170 Pordenone, Italy; 9Department of Precision Medicine, University of Study of Campania “L. Vanvitelli”, 80131 Naples, Italy; 10Experimental and Clinical Pharmacology Unit, Centro di Riferimento Oncologico di Aviano (CRO) IRCCS, Via Franco Gallini 2, 33081 Aviano, Italy

**Keywords:** complementary, alternative, medicine, cancer, chemotherapy, interactions, synergism

## Abstract

**Simple Summary:**

Complementary and alternative medicine (CAM) therapies include a wide range of procedures and products that are often used by cancer patients to directly combat cancer and to protect normal cells from the toxic effects of conventional therapies. Most often, their use is self-prescribed based on a collection of scattered information from websites and advice from relatives or friends. In this paper, we examined the potential known benefits and harms associated with the most commonly used alternative treatments to provide a practical guide for caregivers and patients.

**Abstract:**

CAM is used by about 40% of cancer patients in Western Countries, with peaks of 80% for breast cancer patients. Cancer patients use CAM to boost immune function, to control cancer symptoms and treatment-related side effects, and to improve health-related quality of life (HR-QoL) and survival. Unfortunately, self-prescription of natural remedies in cancer patients can lead to unexpected toxicities and can reduce the effectiveness of cancer therapy. Although CAM usually refers to all the “natural or organic” products/methods that are generally considered less toxic, there are concerns about drug interactions, especially in patients participating in clinical trials with experimental agents. Despite the claims of the promising and potential benefits made by prescribers, many CAMs lack clear scientific evidence of their safety and efficacy. Given the widespread use of CAM—both clearly declared and overt—in this review, we focused on the most important known data on the risk of interactions between biologics and oncology drugs with the goal of opening up CAM in accordance with the meaning of integrative medicine.

## 1. Introduction

Complementary and alternative medicine (CAM) is used by about 40% of cancer patients in Western Countries [1], with peaks of 80% for breast cancer (BC) [2]. Cancer patients resort to the use of CAM for various reasons, including boosting immune function, controlling cancer symptoms and treatment-related side effects, reducing the risk of cancer recurrence, improving health-related quality of life (HR-QoL), and ultimately survival [2,3]. Unfortunately, cancer patients have been known to self-prescribe natural remedies [4], which, in some cases, carries the risk of unexpected toxicities and anticancer drug (ACD) failure.

According to the National Center for Complementary and Integrative Health (NCCIH) in the USA, CAM therapies are divided into five categories (Figure 1) and include a wide range of practices and products that are either biological (e.g., herbs or botanicals, vitamins, minerals, probiotics, homeopathic products, and Chinese herbal remedies) or non-biological (e.g., prayer, meditation, music therapy, and yoga). The division of these practices and products in the respective CAM categories is displayed in Table 1.

These interventions are defined as “alternative” when used instead of traditional medicine (TM) and “complementary” when used in combination with TM [4,5,6]. In addition, the complementary approach opens a new scenario called “integrative medicine” [2].

Considering the scarce or absent known interactions between non-biological CAMs and ACDs, this paper focused on biological CAMs. 

Biological molecules, including green tea, curcumin, quercetin, coenzyme Q10, and medicinal mushrooms, which have been investigated as single agents mainly in cell cultures and preclinical studies, show antitumor activity [7,8,9]. However, the metabolic pathways and mechanisms of action are still unclear when the above compounds are ingested in the diet. In addition, the mixture of these molecules in the diet is believed to have a stronger anticancer effect than the administration of a single isolated compound [4]. Although CAM usually refers to the totality of “natural or biological” products/methods that are considered less toxic overall, there are concerns that CAM may have interactions with drugs, especially in patients participating in clinical trials with experimental agents [6]. CAMs may be responsible for serious adverse events (AEs), such as those shown in Table 2 [7,8,9,10,11,12]. 

An assessment of phase I clinical trials suggested that up to 88% of patients were using CAMs [13]. Some CAM treatments have been carefully tested and have generally been shown to be safe and effective. These include acupuncture, yoga, meditation, some vitamins, nutritional supplements, and herbal remedies, to name a few. Conversely, others do not work, may be harmful (Laetrile), or could negatively interact (St. John’s wort) with ACDs [14]. In addition, adherence to some diets can undoubtedly have beneficial effects [15], while others should be taken with caution and under medical supervision [16]. Despite the claims of the promising and potential benefits made by CAM practitioners, many CAMs lack clear scientific evidence of their safety and efficacy. According to the English-language literature and NCI-NCCIH institution, the main role of most CAMs is to provide supportive care with the primary goal of HR-QoL improvement during and after ACD treatment [17,18,19,20].

Table 3 [7,8,9,16,21,22,23,24,25,26,27,28,29,30,31,32,33,34,35,36,37,38,39,40,41] lists the best-known, studied, and used CAMs whose main recognized role is to improve HR-QoL, although there is conflicting evidence that CAM improves HR-QoL and symptoms [21,22]. 

Some studies suggest that the use of CAM is associated with worse overall survival (OS) in patients with nonmetastatic cancers [21,22,41], suggesting an adverse effect in terms of toxicity, HR-QoL, and survival [42,43]. The worsening effect on survival seems particularly related to the higher refusal of standard treatments in patients using CAMs, as displayed in the Johnson et al. study [42]. Specifically, the study by Burtsein et al. demonstrated greater psychosocial distress and worse HR-QoL due to the use of CAM in patients who received standard treatment for early breast cancer [41]. Compared to this study, the most up-to-date scientific information on the use of CAMs and their role (supportive care, anticancer effect, toxicity) is currently available. In contrast, in a recent study by Connor Wells et al., no worsening of cancer-specific outcomes was observed in patients using CAM, particularly in patients with lung cancer [44], although this result should be interpreted with caution given the retrospective and ad hoc nature of the study.

Some CAMs that are able to improve the HR-QoL of cancer patients may have an indirect anticancer effect from attenuating side effects. Thanks to this beneficial effect, cancer patients might adhere better to ACDs [3,45].

Considering these discrepancies in outcomes, especially in antitumor activity and survival with the use of CAM, we strongly recommend a multidisciplinary and controlled approach for cancer patients treated with CAM or wishing to follow an integrated therapeutic pathway.

The increasing blending of traditional and alternative medicine requires practical guidelines that are useful to clinicians and patients for optimizing the balance between benefits and potential risks. Below, we described a suggested strategy for cancer patients using CAM to avoid potential interactions between CAM and ACDs while enhancing their synergistic effects.

## 2. Methods

A literature search was conducted without language restriction using PubMed and Scopus databases as well as websites related to drug interaction checkers. The inclusion criteria for the assessment of herb–drug interaction were as follows: articles and databases (i.e., http://reference.medscape.com/drug-interactionchecker (accessed on 18 August 2022) on the primary data of any potential herb–drug interaction; studies including cancer patients in clinical (inclusive of all ages and health statuses) and preclinical (in vivo, in vitro, in silico, and assay-based papers) studies; articles investigating an herbal leaf as an intervention in all types of formulations, including that of a raw plant, extract, juice, tablet, capsule, powder, or syrup as a single herb. Exclusion criteria included review articles or reports on secondary data; articles that investigated isolated compounds as interventions, including herbal/nutraceutical-leaf-derived compounds; articles that investigated mixture formulations that contain a leaf as one of their components, along with other active ingredients, as the main intervention (this does not refer to the herb/drug in which potential for interaction was investigated); and articles that investigated single plant parts apart from leaves.

## 3. Drug–Herbal-Product Association: Effects on Pharmacokinetics and Pharmacodynamics

It is a widely held belief that the association between natural products and ACDs allows cancer patients to limit the numerous side effects of a burdensome therapy such as chemotherapy while reaping the benefits that come with it. However, this type of association carries risks that exceed the actual benefits, mainly due to the interaction between the drug and the natural product, which alters the therapeutic window of the drug used in the treatment and promotes the occurrence of toxicity. Figure 2 summarizes the various possible mechanisms of drug–herb interaction [46]. Drug–herb interactions may result in the altered pharmacokinetics of one or both active agents and/or altered pharmacodynamics, affecting their efficacy, especially the positive outcome of drug therapy, which remains as the primary goal. [47]. Metabolic interactions may alter the amount of the drug that reaches the target site, resulting in either therapeutic inefficacy or even toxicity due to an overdose, as is the case, for example, with concomitant use of pineapple with paclitaxel and of Essiac or ginger with sorafenib, docetaxel, or bortezomib, which carries an increased risk of an overdose [10,48]. The co-administration of phytochemicals with the cytochrome system is not recommended and may be harmful [49,50]. The possible induction/inhibition of CYPs that metabolize ACDs is one of the most common changes in drug–natural-product interactions [47]. Therefore, the study of the interactions of natural products with drug-metabolizing enzymes is crucial for their safe concomitant use. This recommendation applies to ACDs and all drugs, especially those with narrow therapeutic windows. 

In vitro and in vivo data on herbal products allow the possibility to recommend the use of a natural product rather than another in association with ACDs [51]. For example, it has been shown that Zingiber Officinalis inhibits CYP2C9 [52] as well as that Curcuma Longa and Panax Ginseng, respectively, inhibit cytochromes CYP1A1, 1A2, and 2B1 and CYP1A1, 1A2 and 1B1 [53,54]. Table 4 summarizes the main herbal products used as adjuvants in pharmacological therapies, the cytochrome isoforms influenced by them (see also Figure 3), and the possible interactions with ACDs. The concomitant use of ACDs and natural compounds is recommended following the assessment of their metabolism. Although metabolism is a key point to consider in this context, it is not the only one. As an example, the concomitant use of natural products may influence the intestinal barrier, altering either the absorption of the drugs or their excretion via the renal system, thus modifying their pharmacokinetic profile [55]. 

Natural products can affect the activity of transport proteins such as P-glycoprotein (P-gp) and thus alter urinary and hepatic excretion of many drugs [62]. A comprehensive understanding of the ADME (administration–distribution–metabolism–excretion)-altering mechanisms of the drug, mediated by the coadministered natural products, is a key factor for safe and efficacious prescriptions. On the other hand, it should be emphasized that natural products or individual isolated secondary metabolites can be adjuvant in various pathologies, reducing the side effects of the use of ACDs or synergizing their action. Prebiotics, probiotics, fungi, and plants should no longer be considered as single dietary supplements but as potential adjuvants in conventional drug therapies: their use can be exploited to reduce the chemoresistance phenomena, enhance the cytotoxic effects of ACDs, or reduce cellular damage caused by ACD use [63]. For example, the most common gastrointestinal side effects of ACD use, nausea and vomiting, can be alleviated by red ginseng and ginger (6-gingerol), as reported in two studies on patients with epithelial ovarian cancer in treatment with platinum-based therapy and in patients with solid tumors receiving highly emetogenic ACD [64,65]. Both the hepatotoxicity associated with radioembolization and the nephrotoxicity induced by the use of ACDs, such as methotrexate, doxorubicin, and 5-fluorouracil (5-FU), are relevant side effects that must be considered [66]. Silymarin, a bioactive compound found in *Silybum marianum*, at a daily dose of 160 to 600 mg has shown beneficial effects in patients with mucosal and liver damage due to radiation therapy. Lycopene as an adjuvant in therapy with cisplatin for the treatment of head and neck tumours has reduced the nephrotoxicity resulting from the use of platinum compounds [36,67,68]. The cardiotoxicity induced by doxorubicin, anthracyclines, and other ACDs is due to the imbalanced effects of energy metabolism [69]. *Platycodon grandiflorum* seems to have a cardioprotective effect on breast cancer patients treated with anthracyclines [70]. In addition to the use of natural products to reduce the side effects of ACDs, there is also the synergistic use of herbal remedies to improve the therapeutic response to a particular drug. Curcumin, for example, has been used in Asian countries for decades due to its recognized anti-inflammatory, antioxidant, and immunostimulant properties [26]. A phase II study demonstrated the synergistic effect between a phospholipid-derivatized curcumin extract and gemcitabine, combining the antiproliferative activity of curcumin with the anticancer effect of the antiblastic agent in patients with pancreatic cancer [27,71]. Particularly, curcumin improved the efficacy of gemcitabine, and their association was well-tolerated in the patients with locally advanced and metastatic pancreatic cancer in the trial [71]. Another example of a synergistic effect is provided by medicinal mushrooms, which are considered an excellent source of potential biomolecules with antimicrobial, anti-inflammatory and antitumor properties [34]. In particular, two medicinal mushrooms belonging to *Polyporaceae*, namely lingzhi (reishi) and yunzhi, are used as adjuvants in cancer therapy not only in the East but also in Canada, the United States, and Europe. At the clinical level, yunzhi is coadministered with ACDs such as 5 FU, tegafur/uracil, and mitomycin, while linghzi is coadministered with cisplatin, paclitaxel, and 5 FU in most of the clinical trials that were analyzed [35]. From the analysis of 77 clinical trials, combining these two fungi with ACDs could improve the survival rates and HR -QoL of patients under cancer treatment [35]. The disclosure of the potential benefits of mushrooms and the assessment of their metabolism are crucial for the safe combination with ACDs [72].

## 4. The Herb–Cancer-Drug Interaction Mediated by Cytochrome P450 (CYP) 

Phase I enzymes, CYPs, are responsible for detoxification and protect human homeostasis from xenobiotics, including drugs and carcinogenic agents. Detoxification also occurs via conjugation reactions [49,50].

The results of pharmacodynamic interactions can be either additive, as in the case of Aloe vera which inhibits CYP1A1 and enhances cisplatin activities, or antagonistic, as in the case of Curcuma which reduces the efficacy of tamoxifen [73].

The induction/inhibition of phase II conjugation enzymes, such as UDP-glucuronosyltransferase isoform 1A1 (UGT1A1) and the glutathione S-transferase family (GSTP1, GSTM, GST), is responsible for the alteration of the detoxification of many carcinogens and ACDs. 

In addition, the pharmacokinetics of many drugs and supplements could also be altered by the induction/inhibition of transporter proteins such as P-gp [74]. P-gps are expressed in the intestine, liver, and kidney and play an important role in the absorption, distribution, or excretion of drugs.

In light of previously reported information, developed countries recognize and value the importance of research on CAM, including herbal medicines [6].

The Office of Dietary Supplements in the USA is an agency that has established research funding and that emphasizes the relevance and clinical importance of pursuing research on drug–herb interactions.

The CYP1A subfamily (CYP1A1 and CYP1A2) plays a key role in the metabolism of many important classes of environmental carcinogens: polycyclic aromatic hydrocarbons (e.g., from cigarettes) and arylamines. Many clinically important drugs are metabolized by this specific isoenzyme, such as platin derivates, caffeine, theophylline, verapamil, and clozapine. Curcuma has a strong interaction with the CYP1A enzymes (Figure 3). In addition, cruciferous vegetables, such as cauliflower, sprouts, and cabbage, are strong CYP1A1 and CYP1A2 inductors [75]. Resveratrol has been demonstrated to be an aryl hydrocarbon receptor antagonist which inhibits the induction of the CYP1A1 enzyme [61], and it is used in cancer patients who receive ACDs [17]. Caffeine and tizanidines are used as inhibitor probes for drug–herb interaction studies. The most important genetic variant affecting CYP1A2 activity is *CYP1A2*F* (−163C > A) rs7625551 in 5′UTR.

CYP2C8 is a partial metabolizer enzyme in both Valerian (*Valeriana officinalis*) and Saw palmetto (*Serenoa repens*) [60]. CYP2C8 is the main enzyme responsible for the metabolism of the anticancer drugs taxanes [76].

The most important and noteworthy genetic variant of diagnostics is the missense polymorphism *CYP2C8*3* rs11572080 (c.416G>A, R139K).

CYP2C9 is responsible for the metabolism of fukinolic acid and cimicifugic acids A and B present in a black cohosh (*cimicifuga racemose*) [49,50] as well as numerous drugs, including warfarin. Relevant genetic variants affecting CYP2C9 enzyme activity are *CYP2C9*2* (c.430C > T, R144C) rs1799853 and *CYP2C9*3* (c.1075A > C, I359L) rs1057910.

CYP2C19 metabolizes (−)-epigallocatechin-3-gallate present in green tea (*camelia Sinensis*) [49,50] as well as drugs including clopidogrel and pantoprazole. The most common variant affecting enzyme expression is *CYP2C19*17* (−806C > T) rs12248560 located in 5′UTR.

CYP2D6 is responsible for the metabolism of more than 30 clinically important drugs, such as metoprolol and several other β-blockers, antiarrhythmics, antidepressants, neuroleptics, morphine, and ACDs, including tamoxifen [77]. The CYP2D6 enzyme is highly polymorphic (*CYP2D6*3* 2459delA frameshift rs35742686, *CYP2D6*4* 1846G > A splicing rs3892097, *CYP2D6*10* c.100C > T P35S rs1065852, *CYP2D6*xN* copy number variation, etc.), and the variants are responsible for interindividual differences in response to xenobiotics. Among drugs, tamoxifen and its active metabolite endoxifen deserve special attention due to the risk of recurrence associated with the 10% of poor metabolizer patients. In addition, cancer patients prescribed tamoxifen treatment should be aware when assuming concomitant herbal supplements, including curcumin, that alter CYP2D6 activity [28].

CYP3A4 is involved in the metabolism of about 60% of prescribed drugs and 50% of nutraceuticals [78]. CYP3A4 is expressed in many organs, including the gut and liver. CYP3A4 plays an important role in the oxidation of many xenobiotics, including polycyclic aromatic hydrocarbons, heterocyclic amines, aflatoxin B1, and nitrosamines. In addition, it is involved in the oxidation of many hormones, such as testosterone and estrogen, and their metabolites have been shown to generate carcinogen DNA adducts [79]. The activity of CYP3A4 can be influenced by the inhibition or induction of herbs and supplements. Resveratrol as well as other flavonoids (quercetin, naringenin) inhibit CYP3A4 in a time- and NADPH-dependent manner [61].

Among herbal remedies metabolized by CYP3A4, there are ginkgo biloba, echinacea, St. John’s worth, curcuma, and terpenes (essential oil). The interaction between these supplements and ACDs is extensively reported in the literature [80]. 

## 5. Special Consideration for Elderly and HIV-Positive Cancer Patients

Many cancer patients older than 65 years of age take multiple medications and herbal supplements, and they usually do not inform their physicians [5]. This can potentially increase sensitivity to certain drugs and affect the metabolism of many others, such as biguanides [81]. In addition, there are few clinical studies addressing herb–drug interactions in elderly patients [82]. In one clinical study, the elderly were found to be susceptible to age-related changes in CYP sensitivity to herbal supplements. For example, Gurley et al. reported that ginseng, milk thistle, and echinacea inhibited CYP2D6 in elderly subjects, whereas no/low inhibition was observed in younger subjects [83]. For this reason, the self-administration of herbal supplements in elderly patients in concomitant polypharmacy therapy should be strongly discouraged. In addition, in “frail” elderly cancer patients who receive ACDs, self-administration of supplements must be avoided [84]. Like the elderly, people living with HIV (PLWH) [85,86,87] represent a special subgroup of patients for whom the risk of drug–drug interactions (DDI) between antiretroviral therapy (ART), ACDs, and natural products must be carefully evaluated [88,89]. It is well known, that in PLWH patients, polypharmacy therapy is frequent, mainly in those with cancer, disease, or comorbidities related to HIV infection. Nowadays, the availability of more than 20 approved antiretrovirals drugs and the possibility of individual genotype profiling allow the development of protocols that minimize the potential DDIs and improve agreement between ART, ACDs, and CAM treatments [89,90].

## 6. Drug–Drug Interaction Risk Assessment in Cancer CAM Users

A drug–drug interaction is defined as a reaction between two (or more) drugs or between a drug and food or supplements. Such interactions can affect the pharmacokinetics and the pharmacodynamics of drugs and may possibly cause both unwanted side effects and drug activity impairment or improvement [84,91]. Among factors that may influence the risk of DDI, old age and multidrug use are surely crucial as well as chronic diseases. Typically, patients diagnosed with cancer already have at least one chronic disease and, therefore, are administered one or more therapies [92]. Systemic ACD therapy itself is usually composed of multiple antiblastic drugs or radiotherapy, and the management of the more common ACD side effects is usually handled with several drugs to relieve symptoms. It is, therefore, crucial to consider all the ongoing therapies of a patient in order to evaluate the potential interactions that could influence the effectiveness of ACDs or even the development of severe adverse reactions possibly causing hospitalization or therapy discontinuation [93]. Among the substances to be considered for DDI, special attention should be paid to biologically based CAM, mainly because of the low awareness of their potential harm among physicians and the low propensity of patients to disclose their use to physicians [94]. Currently, different useful online tools that can check potential DDIs in cancer patients are available, and their databases are also integrated with several commonly used biologic CAMs. Despite the availability of DDI checker programs, among which the most common are Lexicomp [95], Medscape [96], and Micromedex [97], their use is very limited in clinical practice. Such programs are able to analyze a medication list and to categorize the risk of drug-pair interactions into five categories: severe, major, moderate, minor, and no known interaction (Table 5). When a cancer patient relies on the specialist to decide on the appropriate cancer care, it would be useful if the patient’s medical history were supplemented by an assessment of the risk of interactions both between the drugs and supplements taken by the patient and in relation to the ACDs to be administered to the patient. In addition, if the physician decides to prescribe integrative therapy to a cancer patient to aid in his physical recovery, to alleviate the side effects of ACDs, or to improve his HR-QoL, it is necessary to verify the absence of interactions between the administered medications and the biological CAM [98]. This would not only make their use safer, but it would help both to disseminate the potential harm of biological CAMs and to avoid patients’ reticence to inform physicians about self-prescribed supplements. Furthermore, if a patient experiences unexpected adverse reactions based on the prescribed therapy, the abovementioned programs should be used to perform a causality assessment of the toxicity, thus enabling therapy adjustments or changes. Despite the usefulness of these programs, several challenges still need to be overcome for their use to become routine in clinical practice: first, the databases need to be integrated with all biological CAMs; hospitals should obtain a license for at least one interaction testing program; and physicians should receive courses on how to use these programs. Finally, interdisciplinary networks between physicians and pharmacists/pharmacologists should be promoted in order to guarantee effective pharmacovigilance activity (Figure 4) [99]. 

## 7. Discussion

The use of natural products as remedies for diseases has very ancient origins, as evidenced by the discovery of references to the healing properties of *Aloe* in the Bible.

Recent data indicate that CAM is used to treat a wide range of late-life health conditions, especially chronic or long-term conditions such as arthritis and pain, diabetes, hypertension, depression, anxiety, and sleeps disorders [4]. 

In addition to these specific health disorders, CAM has also been shown to be used by cancer patients to manage the ACD-related side effects and to improve HR-QoL and survival. 

The use of CAM in a cancer setting is currently controversial due to the use of a wide range of “natural” products [41,100,101,102] that are very often self-prescribed or suggested outside of an integrative medical approach and due to the lack of a sufficient number of randomized clinical trials assessing their efficacy and synergism with ACDs. At the same time, another barrier to their implementation in clinical practice is the lack of adequate knowledge of CAM by physicians, making this scenario even more confusing. In the last few years, based on NCI-NCCIH guidelines, well-designed clinical trials and scientific disclosures on CAM have allowed us to take advantage of more clinical tools to evaluate the correct use of natural products, supporting cancer patients with the aim of improving their HR-QoL and possibly their OS [2]. In addition, with the introduction of several herbal products in the databases of DDI checker programs, we have the opportunity to calculate the risk of DDIs between ACDs and natural products [95,96,97,103]. This allows reduced AEs, possibly caused by their interactions and, at the same time, identifies the potential benefits and synergic activity between ACDs and natural products [67,68]. Despite the absence of unequivocal causal relationships by now, a great amount of clinically relevant pharmacokinetic interactions have been detected throughout the years. Many of them seem to involve whether the inhibition or the induction of several isoforms of CYP was determined by natural products. 

In addition, CYP induction/inhibition is also influenced by both food intake, including that of grapefruit, ginger, and green tea, and polymorphisms affecting the functionality and expression of these phase I enzymes [104]. All these overlapping factors make it difficult to determine the phenotypic response to drug treatments and the possible risk of interactions between drugs and herbal supplements. 

The role of these enzymes in the metabolic and elimination process of many ACDs is crucial, which could lead to serious toxicological implication if ACDs were concomitantly administered with CAM products without a previous rigorous DDI assessment. At present, the use of biological CAMs by cancer patients has been increasing [105]; thus, it is necessary for oncologists to collect information about them in the patients’ drug history in order to evaluate and discuss potential benefits or disadvantages of their association with ACDs. Importantly, most cancer patients use self-prescribed biological CAMs [6]. In Italy, this growing phenomenon of self-care and the refusal of TM is bound to be permanent rather than passing [106], even though it has been established that the use of “unconventional” treatments for curable cancers not associated with conventional ACDs could increase mortality rate [107]. Johnson et al. recently confirmed this theory, observing that CAM users showed a poorer 5-year survival rate and a higher risk of death when compared with patients not using CAMs, especially due to the refusal of TM (ACD, surgery, radiotherapy) outlined in patients using CAMs [43]. 

Despite the existence of DDI checker programs, including Lexicomp and Medscape, and the increasing disclosure of the potential harm of biological CAMs, their implementation into routine practice via an integrative approach is still a challenge. Therefore, oncology departments should consider defining guidelines aimed at (a) identifying cancer patients at risk of self-prescribing CAMs; (b) improving physician-patient communication by adopting an open attitude toward biological CAMs; (c) taking an accurate pharmacological history; (d) highlighting the risk of DDIs between CAMs and ACDs and polypharmacy; (e) suggesting the safe use of biological CAMs capable of improving the HR-QoL of cancer patients; (f) and, finally, adequately informing the patients about the risks and benefits of the use of CAM as recently outlined by an Italian study [108]. 

## 8. Conclusions

In conclusion, complementary or alternative medicine topics are of great interest to cancer patients. It is clear that the combination of CAM and ACD treatments could be dangerous for cancer patients, especially if not prescribed by a multidisciplinary and experienced team dedicated to this issue. In our opinion, safe and useful complementary therapies should be integrated into regular cancer treatment to improve HR-QoL and the clinical outcomes of patients. Thanks to DDI checker programs and excellent educational materials from reputable sources, we have the opportunity to recommend the right integrative approach to cancer patients, especially those being treated with anticancer drugs. In the near future, we would like to see a robust integrative oncology program as a cancer treatment in hospitals that is available to physicians and patients.

## Figures and Tables

**Figure 1 cancers-14-05203-f001:**
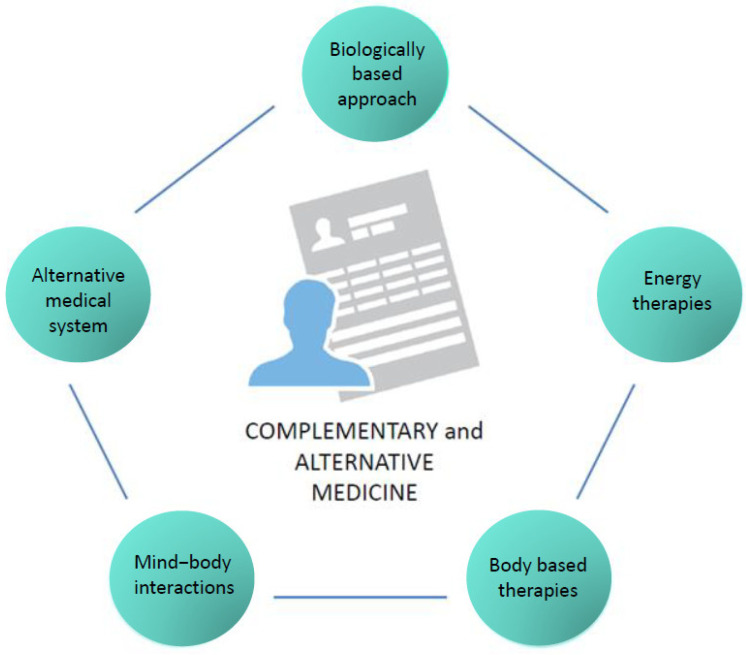
CAM categories.

**Figure 2 cancers-14-05203-f002:**
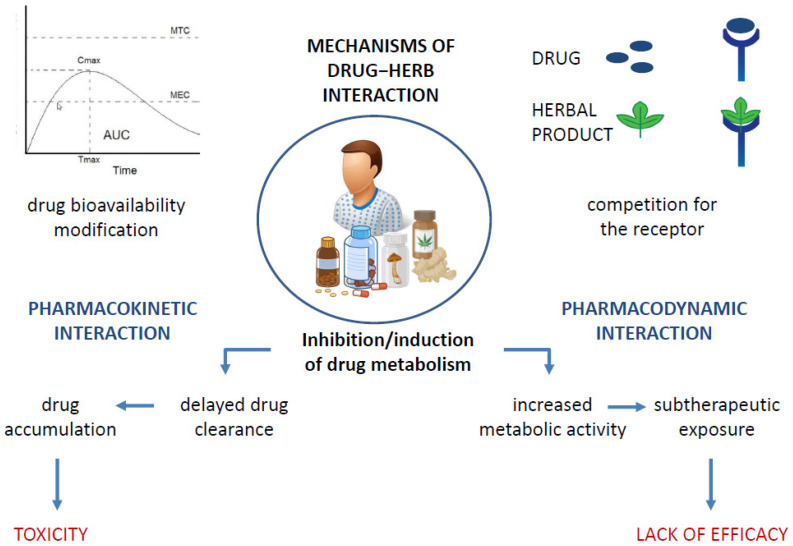
Possible mechanisms of drug−herb interactions, leading to toxicity development or lack of drug efficacy.

**Figure 3 cancers-14-05203-f003:**
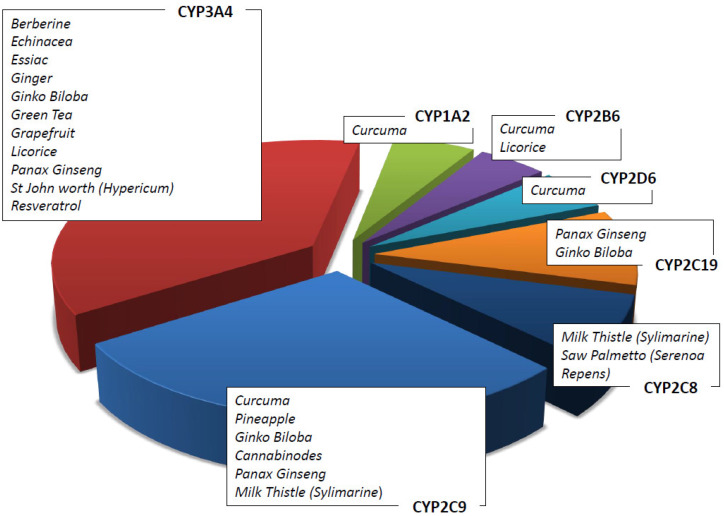
Herbs/nutraceuticals commonly used as CAMs that induce and/or inhibit different CYP subfamilies.

**Figure 4 cancers-14-05203-f004:**
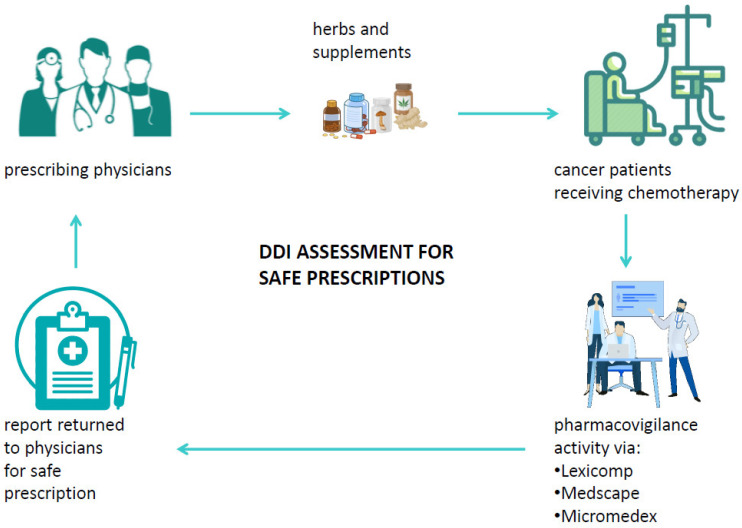
Suitable pharmacovigilance activity via DDI checker programs in cancer patients’ journey.

**Table 1 cancers-14-05203-t001:** Different CAM therapies and respective categories.

CAM Categories	Mind–Body Therapies	Biologically Based Practices	Manipulative & Body-Based Practices	Energy Therapies	Alternative Medical Systems
Meditation	✓	✕	✕	✕	✕
Biofeedback	✓	✕	✕	✕	✕
Hypnosis	✓	✕	✕	✕	✕
Yoga	✓	✕	✕	✕	✕
Tai Chi	✓	✕	✕	✕	✕
Vitamins	✕	✓	✕	✕	✕
Dietary Supplements	✕	✓	✕	✕	✕
Botanicals	✕	✓	✕	✕	✕
Herbs	✕	✓	✕	✕	✕
Special foods or diets	✕	✓	✕	✕	✕
Massage	✕	✕	✓	✕	✕
Chiropractic therapy	✕	✕	✓	✕	✕
Reflexology	✕	✕	✓	✕	✕
Reiki	✕	✕	✕	✓	✕
Therapeutic touch	✕	✕	✕	✓	✕
Ayurvedic medicine	✕	✕	✕	✕	✓
TCM	✕	✕	✕	✕	✓
Homeopathy	✕	✕	✕	✕	✓
Neuropathic medicine	✕	✕	✕	✕	✓

Abbreviations: CAM is complementary and alternative medicine; TCM is traditional Chinese medicine.

**Table 2 cancers-14-05203-t002:** Examples of most serious and frequent AEs caused by CAMs.

CAMs	Adverse Events
Aloe vera [7]	Diarrhea, hepatitis
Cannabis [8]	Emesis, dizziness
Green tea [9,10]	Emesis, insomnia, diarrhea
Gingko [10]	Emesis, migraine
Echinacea [10]	Hypersensitivity reactions
Ginseng [11]	Diarrhea, migraine, hypertension, nausea
St. John’s wort [12]	Nausea, hypersensitivity reactions

**Table 3 cancers-14-05203-t003:** Clinical effects of CAM herbs/nutraceuticals.

CAM Activity	Anticancer	Immunomodulation	Control AEs	Improvement HR-QoL	Synergic Activity	Serious AEs	Worsening Outcome
Acupuncture	✕	✓ *	✓✓✓	✓✓✓	✕	✕	✕
Aloe vera (gel)	✕	✕	✓✓✓	✓✓✓	✕	✕	✕
Aloe vera [7]	✓✓	✓✓	✕	✕	✕	✓✓✓	✓✓✓
Aromatherapy	✕	✓	✓✓✓	✓✓✓	✕	✕	✕
Cannabis Cannabinoids [8,23,24]	✕	✕	✓✓✓	✓	✕	✕	✕
Coenzyme Q10 [21,22,23,24,25]	✓	✓✓✓	✓✓✓	✓✓✓	✕	✕	✕
Curcumin [26,27,28,29,30,31]	✓✓✓	✕	✓	✓✓✓	✓✓✓	✕	✕
Essiac [32]	✓	✕	✕	✕	✓	✕	✕
Food/dietary Supplements [16,33]	✕	✕	✓✓	✓✓	✓	✓✓	✓
Green Tea [9]	✓✓✓	✕	✓✓✓	✓✓✓	✕	✕	✕
IV Vitamin C [34]	✕	✕	✓✓	✓✓	✕	✕	✕
Laetrile	✕	✕	✕	✕	✕	✓	✓
Medicinal Mushrooms [22,35,36]	✕	✓✓✓	✓	✓	✕	✕	✕
Milk Thistle [37]	✕	✕	✓	✓	✕	✕	✕
Mistletoe Extracts [38]	✓✓✓	✓✓✓	✓✓✓	✓✓✓	✓	✕	✕
Zeolite [39,40]	✕	✓✓	✓✓✓	✕	✕	✕	✕

Legend: HR-QoL is the health-related quality of life; AEs are adverse events; most studies were laboratory/animal/preclinical studies; * means human/clinical studies; and serious AEs are often dose-dependent. ✓✓✓ means probably efficacious (data from RCTs); ✓✓ means might be efficacious (data from RCTs with smaller samples; ✓ means could be effective (single-arm studies); and ✕ means no sufficient data.

**Table 4 cancers-14-05203-t004:** Most common herbs/nutraceuticals used in cancer patients and their possible interactions with drugs.

Agents	Effect on Metabolic Pathway	Interaction with Anticancer Drugs
Aloe Vera	CYP1A1 and CYP1A2 downregulation	Antiproliferative effects; Cisplatin enhancement [7]
Pineapple (Bromeline)	CYP2C9 inhibition	Risk of overdosage with paclitaxel [48]
Berberine (glodenseal)	CYP3A4 inhibition	Risk of overdosage with bortezomib and dasatinib; influencing pioglitazone hypoglycemic drugs [56]
Curcuma	CYP1A2, CYP2B6, CYP2C9, CYP2D6 weak inhibition	Risk of overdosage with Bendamustine; increased tacrolimus level; risk of inefficacy of prodrugs (Cyclophosphamide, Tamoxifen, etc.) [57]
Cannabinoids	CYP2C9 induction	Risk of overdosage with temozolomide and prodrugs (Cyclophosphamide, Tamoxifen, etc.) [58]
Echinacea	CYP3A4 induction	Improved pharmacokinetics (weak) of Cyclophosphamide, dasatinib, docetaxel, erlotinib, imatinib, sorafenib, and vinca alkaloids [10]
Essiac *	CYP3A4 inhibition	Risk of overdosage with bortezomib, dasatinib, docetaxel, erlotinib, imatinib, sorafenib, and vinca alkaloids [10]
Ginger	CYP3A4 inhibition	As for Essiac [10]
Gingko Biloba	CYP2C19, CYP2C9, CYP3A4, UGT1A1 (in vitro), P-gp, OATP	As for Essiac [10]
Green Tea	CYP3A4 inhibition	As for Essiac [10]
Grape Fruit (naringenine, quercetine, bergamottine)	CYP3A4 OATP1A2 OATP2B1 inhibition	As for Essiac [10]
Licorice	CYP2B6, CYP3A4 weak inhibition	As for Essiac (weak) [10]
Milk thistle (Silymarin)	UGTA1A inhibition, CYP2C8, CYP2C9 weak inhibition	Risk of overdosage with Irinotecan, cyclophosphamide, and paclitaxel [59]
Panax Ginseng	CYP2C19, CYP2C9, CYP3A4	Imatinib [11]
St. John’s wort (Hypericum)	CYP3A4 induction. P-gp induction	Improved pharmacokinetics of Cyclophosphamide, dasatinib, docetaxel, erlotinib, imatinib, sorafenib, and vinca alkaloids [12]
Saw palmetto (serenoa Repens)	CYP2C8, partial inhibition	Improve pharmacokinetics of Taxanes [60]
Resveratrol	CYP1A1, CYP3A4 inhibition	Decreased activation of carcinogenic agents [17,61]

Source: http://reference.medscape.com/drug-interactionchecker (accessed on 18 August 2022). Legend: Cytochrome P450 is CYP; Organic Anion Transport protein is OATP; P-glycoprotein is P-gp; and Essiac * is an herbal mixture patented as an anticancer therapy in 1920 by Rene Caisse in Canada.

**Table 5 cancers-14-05203-t005:** Categorization of DDI severity.

Category	Lexicomp (Need Subscription)	Medscape (Free Use)	Micromedex (Need Subscription)
*Severe*	Avoid combination	Contraindicated	Contraindicated
*Major*	Consider therapy modification	Serious-use alternative	Major
*Moderate*	Monitor therapy	Monitor closely	Moderate
*Minor*	No action needed	Minor	Minor
*No interaction*	No known interaction	No interaction found	Unknown

## Data Availability

No new data were created or analyzed in this study. Data sharing is not applicable to this article.

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
