# Peer review of "Evaluation of Concomitant Use of Anticancer Drugs and Herbal Products: From Interactions to Synergic Activity"

_cancers, 2022, doi:10.3390/cancers14215203_

Round 1
Reviewer 1 Report
How is - use of CAM associated with poor outcomes, in non metastatic cancers? can the authors explain the reason for it?
ref 26 is a review- but it has been refernced as saying that Curcumin, for example, has been used for decades for its recognized 257
anti-inflammatory, antioxidant, and immunostimulant properties,which is not true.
ref number 27 is also a small phase 2 study with 17 pts, for which 11 continued drug, and 55% had progressive disease. which means its very likely due to gemcitabine alone. and there was no placebo comparing study.
refernce 35 is also a review. there are no phase 2/ 3 randomised studies, comparing with standard of care or placebo
Author Response
Dear reviewer,
Thank you for your comments.
In response to point 1, we added the reason for the poorer outcome of non-metastatic patients using CAMs in line 130.
For what concerns ref 26, in the first paragraph of the cited review it is underlined how Curcumin has traditionally been used in Asian countries for its properties. We added “in Asian countries” in line 267 to be more specific.
We would like to point out that this paper is an Opinion Paper (and not a review). We are waiting for the editorial assistant to modify the type of article.
Kind regards.

Reviewer 2 Report
This review would provide the useful information and potential concerns about the concomitant use of anti-cancer drugs and natural products but it is recommended to check some sections and review in a comprehensive manner.
In this review article, authors considerably summarized the possible interactions between anti-cancer drugs and natural products and their side effects. This study would be helpful to give insights about the concomitant use of anti-cancer drugs and CAMs. But I think some sections need more clarification like section-3 with example of drugs. Moreover, I have found following issues that are needed to be addressed.
Line 32 and also throughout the manuscript, grammatical error.
Some figure captions are bold, while some are not, for eg., figure 1 & 2.
Context of figure-1 & 2 ae not understandable and blur.
Caption of figure-3 seems confusing
Line 91 refers combination of ACDs and CAMs into table-2 but table-2 only shows AEs associated with CAMs
Section-3 seems confusing, good to use some example of drug interaction on line 169-172
Some captions of table are small and bold while some are capital; good to be consistent
Line 193, extra space before the concomitant.
Author Response
Dear reviewer,
Thank you for your comments.
We provided to make up for all that you pointed out.
All captions of figures and tables have been uniformed
We tried to clarify the context of figure 1 and 2
We modified the caption of figure 3
We modified the text in line 91
We have added some examples in section 3 (lines 182-184) and made the paragraph more fluent.
We removed the extra space in line 193.
Kind regards

Reviewer 3 Report
In this review, the authors described the categories of complementary or alternative method for cancer treatment. Major concerns are:
1, The authors listed several categories of CAM, but in the main text mainly discussed the herb products. Either the contents should be extended, or the topics and the title of the manuscript can be modified to fit the contents.
2, The title state indicates an evaluation of the CAM, however, the effects of CAM is naturally very hard to evaluate, due to a lack of scientific documentation from the mechanisms to the clinic effects. Plus, I did not see any efficient evaluation method is raised in this manuscript.
Author Response
Dear reviewer,
Thank you for your comments.
We provided to modify the title in “Evaluation of concomitant use of anti-cancer drugs and herbal products: from interactions to synergic activity” and we specified in line 85 the type of CAM the paper focuses on.
We would also like to point out that this paper is an Opinion Paper (not a review). We are waiting for the editorial assistant to modify the type of article.
Kind regards

Round 2
Reviewer 1 Report
good paper, all previous changes incorporated
Reviewer 3 Report
I have no further comments.